# The Effect of Systemic Inflammation and Clinicopathologic Features on Survival in Malignant Pleural Mesothelioma: A Multicenter Analysis

**DOI:** 10.3390/medicina61010144

**Published:** 2025-01-16

**Authors:** Nadiye Sever, Sedat Yıldırım, Ali Fuat Gurbuz, Delyadil Karakaş Kılıç, Esra Zeynelgil, Yunus Emre Altintaş, Berivan Deniz Çimik, Yeşim Ağyol, Ali Kaan Güren, Pınar Erel, Erkam Kocaaslan, Burak Paçacı, Mustafa Alperen Tunç, Abdussamet Çelebi, Nazım Can Demircan, Selver Işık, Rukiye Arıkan, Murat Araz, Serdar Karakaya, Murat Sarı, Osman Köstek, İbrahim Vedat Bayoğlu

**Affiliations:** 1Medical Oncology, Department of Internal Medicine, Marmara University Faculty of Medicine, İstanbul 34854, Turkey; yesimagyol@gmail.com (Y.A.); alikaanguren@gmail.com (A.K.G.); pnarerell@gmail.com (P.E.); erkamkocaaslan@gmail.com (E.K.); drpacaci@gmail.com (B.P.); m.alperen.tunc@gmail.com (M.A.T.); abdussametcelebi@gmail.com (A.Ç.); dr-selver83@hotmail.com (S.I.); dr_rukiyearikan@hotmail.com (R.A.); dr.vebay@gmail.com (İ.V.B.); 2Medical Oncology, Department of Internal Medicine, Kartal Training and Research Hospital, İstanbul 34865, Turkey; rezansedat@hotmail.com (S.Y.); yunusaltintas1688@gmail.com (Y.E.A.); 3Medical Oncology, Department of Internal Medicine, Necmettin Erbakan University, Konya 42090, Turkey; dr.alifuatg@gmail.com (A.F.G.); zaratarum@yahoo.com (M.A.); 4Medical Oncology, Department of Internal Medicine, Dicle University Faculty of Medicine, Diyarbakır 21280, Turkey; karakasdelyadil@gmail.com; 5Medical Oncology, Department of Internal Medicine, Ankara Ataturk Senatorium Training and Research Hospital, Ankara 06290, Turkey; esra23.05@hotmail.com (E.Z.); drserdarkarakaya@gmail.com (S.K.); 6Department of Internal Medicine, Marmara University Faculty of Medicine, İstanbul 34854, Turkey; bdcimik@hotmail.com; 7Medical Oncology, Department of Internal Medicine, Başakşehir Çam and Sakura City Hospital, İstanbul 34480, Turkey; ncdemircan@gmail.com; 8Medical Oncology, Department of Internal Medicine, İstanbul Medipol University, İstanbul 34810, Turkey; drmuratsari@gmail.com (M.S.); osmankostek@yahoo.com (O.K.)

**Keywords:** malignant pleural mesothelioma, C-reactive protein/albumin ratio (CAR), neutrophil/lymphocyte ratio (NLR), epithelioid histology, survival

## Abstract

*Background and Objectives:* Malignant pleural mesothelioma (MPM) is a rare and aggressive malignancy with a poor prognosis. Identifying reliable prognostic factors is crucial for risk stratification and optimizing treatment strategies. This study aimed to evaluate the impact of clinicopathologic factors and systemic inflammatory markers on survival outcomes in patients with MPM. *Materials and Methods:* This retrospective, multicenter study included 217 patients diagnosed with MPM between January 2009 and March 2024. Data on age, gender, histology, disease stage, treatment modalities, and inflammatory markers such as the neutrophil-to-lymphocyte ratio (NLR) and C-reactive protein/albumin ratio (CAR) were collected. Survival outcomes were analyzed using Kaplan–Meier methods, and prognostic factors were evaluated using Cox regression analysis. *Results:* CAR was identified as an independent prognostic factor for both overall survival (OS) and progression-free survival (PFS). Patients with CAR < 0.98 had significantly longer OS (87.0 months vs. 14.0 months, *p* < 0.001) and PFS (17.61 months vs. 8.96 months, *p* = 0.010). While NLR was significant in univariate analysis (OS: 25.0 months for NLR < 2.58 vs. 21.0 months for NLR ≥ 2.58, *p* = 0.040), it did not retain significance in the multivariate model (*p* = 0.180). Epithelioid histology and early-stage disease were strongly associated with improved survival outcomes (OS: 32.0 vs. 11.0 months for epithelioid vs. non-epithelioid histology, *p* < 0.001; 32.0 vs. 12.0 months for early-stage vs. metastatic disease, *p* < 0.001). *Conclusions:* CAR is a strong independent prognostic factor in MPM, reflecting systemic inflammation and nutritional status. Epithelioid histology and early-stage disease are associated with significantly longer survival, underscoring the critical role of early detection in improving patient outcomes.

## 1. Introduction

Malignant pleural mesothelioma (MPM) is a rare and aggressive malignancy arising from mesothelial cells of the pleura, often associated with asbestos exposure [1]. It has three main subtypes, which are epithelioid, sarcomatoid and biphasic. Treatment options include surgery, chemotherapy, and immunotherapy. The prognosis is poor, with a median overall survival (OS) of 9 to 17 months after diagnosis [2,3]. Male gender, non-epithelioid histology, and advanced age are known as poor prognostic factors [4]. Identifying reliable prognostic factors in this challenging disease is of great importance for risk stratification and optimizing treatment approaches.

Chronic inflammation is considered one of the key features of cancer, contributing to tumor initiation, progression, and metastasis [5]. Studies have shown that chronic inflammation is an important factor in determining the prognosis of various types of cancer, including malignant mesothelioma (MM) [6]. The neutrophil/lymphocyte ratio (NLR), platelet/lymphocyte ratio (PLR), and C-reactive protein (CRP)/albumin ratio (CAR), which are markers of systemic inflammatory response, have emerged as potential prognostic indicators in various malignancies [7]. CAR, in particular, has shown promise as a powerful predictor of survival by reflecting systemic inflammation and providing indirect insights into nutritional status [8]. We frequently use these low-cost and easily accessible markers in clinical practice. However, to date, there have been few studies that have investigated the association between patients’ inflammatory markers and nutritional status using standardized assessment tools. In this study, we aimed to evaluate the prognostic effects of systemic inflammation markers on survival in patients with MPM. Our study aims to contribute to the understanding of inflammation-based prognostic models and their potential role in guiding individualized treatment strategies.

## 2. Material and Methods

Patients who were diagnosed with pleural mesothelioma between January 2009 and March 2024 and who were followed up and treated in the oncology outpatient clinic were included in the study. This study was conducted as a multi-center retrospective analysis, including patients whose treatment and follow-up were carried out at multiple independent healthcare institutions. Physicians from these centers contributed clinical data for analysis through a collaborative effort. Ethics committee approval of our study was obtained from the Marmara University Faculty of Medicine Ethics Committee on 22 April 2024 with protocol number 09.2024.500. Age, gender, Eastern Cooperative Oncology Group (ECOG) performance status, tumor histology, stage at diagnosis, presence of surgery, type of surgical intervention if performed (extrapleural pneumonectomy [EPP]/pleurectomy-decortication [P/D]), recurrence status in patients who received curative treatment, treatment regimens used in systemic treatment were examined. Also, total lymphocyte, total neutrophil, platelet counts, hemoglobin, serum albumin, and CRP values at the initial presentation were recorded. The information of the patients was retrospectively scanned from the patient files and the electronic record system of the hospital. The relationship between the data obtained and progression-free survival (PFS) and OS was analyzed. PFS was calculated as the time between the start of systemic therapy and the date of disease progression. OS was expressed as the time from the date of diagnosis to the date of death from any cause or the date of last follow-up for surviving patients.

### 2.1. Definitions and Formulae

All indices were based on the clinical and laboratory parameters from patients’ initial diagnosis. The indices were computed using the following formulae: NLR; absolute neutrophil count (count/mm^3^)/absolute lymphocyte count (count/mm^3^), CAR; CRP (mg/dL)/serum albumin (g/dL).

### 2.2. Statistical Analysis

Data analysis was performed using SPSS 26.0 statistical software. Continuous variables were assessed for normality. Variables with non-normal distributions are presented as median and interquartile range, whereas variables with normal distribution are presented as mean and standard deviation. While categorical variables were analyzed using the Chi-square or Fisher’s exact test. The ideal cut-off value to predict NLR and CAR was calculated by the receiver operating characteristic (ROC) analysis. Survival curves were generated using the Kaplan–Meier method for each subgroup, with 95% confidence intervals (CIs). The log-rank test was used to compare differences in survival between groups. Prognostic factors were examined using univariate analysis, with subsequent examination of factors with a *p*-value of less than 0.5 in the multivariate analysis. Hazard ratios (HRs) for these comparisons were calculated using a Cox proportional hazards model. Statistical significance was established at *p* < 0.05.

## 3. Results

### 3.1. The Study Population’s Demographic and Clinical Characteristics

This study included 217 patients diagnosed with MPM. The median age was 59 years (IQR: 52–68.5). The majority of patients were male (64.5%), and epithelioid histology was the predominant subtype (68.7%), followed by sarcomatoid (17.1%) and biphasic (14.3%) histologies. Asbestos exposure was reported by 58.1% of patients, and tobacco exposure was documented in 49.8%. At diagnosis, most patients presented with stage I-III disease (62.7%), while 37.3% were diagnosed at stage IV. Surgical treatment was performed in 41.9% of the cohort, primarily PD in 73.3% of surgical cases, while 26.7% underwent EPP (Table 1).

### 3.2. Survival Analysis

The median OS time was 18.0 months (95% CI: 22.8–29.9 months). The median PFS time was 12.8 months (95% CI: 12.2–17.9 months).

Epithelioid histology was associated with longer PFS compared to non-epithelioid subtypes (14.78 months vs. 9.75 months, *p* = 0.437), although this did not reach statistical significance. ROC analysis was used to determine the ideal cut-off value for systemic inflammatory markers (Figure 1). CAR was significantly associated with PFS in univariate analysis. Patients with CAR < 0.98 exhibited longer PFS compared to those with CAR ≥ 0.98 (17.61 months vs. 8.96 months, *p* = 0.010). However, NLR did not show a significant association with PFS (*p* = 0.550) (Table 2).

Univariate analysis revealed several clinical and pathological factors significantly associated with OS. Epithelioid histology demonstrated a notable survival advantage over non-epithelioid subtypes (median OS: 32.0 months vs. 11.0 months, *p* < 0.001) (Figure 2). Similarly, patients without metastatic disease at diagnosis exhibited significantly improved OS compared to those with metastases (median OS: 32.0 months vs. 12.0 months, *p* < 0.001). While univariate analysis demonstrated a significant association between surgery and prolonged survival in operated patients (*p* < 0.001), this relationship was not confirmed in the multivariate analysis (*p* = 0.914). The use of immunotherapy in any treatment line was associated with a longer median OS compared to those who did not receive immunotherapy (45.0 months vs. 21.0 months, *p* = 0.016). In terms of systemic inflammatory markers, CAR was significantly associated with OS in both univariate and multivariate analyses (Figure 3). Patients with CAR < 0.98 had a dramatically longer median OS compared to those with CAR ≥ 0.98 (87.0 months vs. 14.0 months, *p* < 0.001). Multivariate analysis confirmed CAR as an independent prognostic factor for OS (HR = 0.17, 95% CI: 0.10–0.28, *p* < 0.001). While the neutrophil-to-lymphocyte ratio (NLR) was significant in univariate analysis (median OS: 25.0 months for NLR < 2.58 vs. 21.0 months for NLR ≥ 2.58, *p* = 0.040), this significance was not retained in the multivariate model (HR = 1.27, 95% CI: 0.89–1.81, *p* = 0.180) (Table 3).

## 4. Discussion

In this study, we examined the impact of clinicopathologic factors and systemic inflammatory markers on survival outcomes in patients with MPM. Among clinicopathologic factors, epithelioid histology and absence of metastatic disease at diagnosis were strongly associated with prolonged OS and PFS, consistent with their established role in MPM prognosis. Furthermore, systemic inflammatory markers, especially CAR, were independent prognostic factors for prolonged survival. CAR was associated with OS and PFS, demonstrating its utility as a biomarker reflecting the combined effects of systemic inflammation and nutritional status. While NLR showed significance in univariate analysis, it did not retain this association in multivariate models, suggesting that its prognostic value may be influenced by other overlapping factors or broader systemic dynamics reflected in composite indices such as CAR.

There is increasing evidence in the literature supporting the association between inflammatory markers and the overall prognosis in a wide variety of cancers [9,10,11]. Hypotheses previously put forth regarding the etiology of malignant mesothelioma suggested that inflammation could be a contributing factor to the disease. Due to the hypothesis implicating long years of inflammation in the etiology of MPM, the focus of the search for a prognostic biomarker has shifted to inflammation markers [12]. A study of 115 MM patients from Turkey investigated the prognostic value of various inflammation markers and found that high pan-immune inflammation and high systemic inflammation response index were associated with worse OS [13]. Takamori et al. reported the CAR was an independent prognostic marker in MPM [14]. In another study of 132 patients from Turkey, CAR was not a predictor of prognosis. According to a one-year survival analysis, there was no statistically significant difference in mean CAR between those who lived less than one year and those who lived more than one year [15]. The results of our study are consistent with previous studies emphasizing the prognostic importance of systemic inflammatory markers in MPM.

CAR demonstrates its prognostic significance in cancer due to its ability to capture both systemic inflammation and nutritional status, two key factors associated with tumor progression and survival [16]. The high levels of CRP, an acute-phase protein, are indicative of a pro-inflammatory state driven by cytokine release (e.g., interleukin-6), which supports tumor growth, angiogenesis, and immune evasion [17]. Concurrently, low albumin reflects poor nutritional reserves and impaired immunity, both linked to worse outcomes. This dual representation of the host’s inflammatory and metabolic state may enhance CAR’s prognostic power compared to single biomarkers like NLR [18]. The interaction between inflammation and nutrition likely explains the superior prognostic value of CAR compared to isolated markers. By integrating these two dimensions, CAR provides a more comprehensive snapshot of the patient’s systemic environment, highlighting its potential as a robust biomarker in oncological settings.

NLR is a biomarker of increasing interest in cancer prognosis as an indicator of systemic inflammation. Studies have shown that high NLR values are associated with poor clinical outcomes, indicating increased inflammation and suppressed immune function [19]. Kao et al. emphasized the importance of NLR in predicting prognosis in MPM [20]. In this study, a one-year survival rate was reported as 60% for NLR < 5 and 26% for NLR ≥ 5. A previous meta-analysis showed that high NLR was identified as an unfavorable prognostic factor in patients with malignant mesothelioma [21]. In a study from Turkey, there was a significant association between NLR and survival [13]. Tural Onur et al. NLR was investigated as a prognostic marker in MPM, and no significant relationship was found between NLR and prognosis [22]. In our study, although NLR was significant in univariate analysis, it failed to maintain its prognostic significance in multivariate analysis. In this context, our findings highlight that while NLR reflects systemic inflammatory responses and demonstrates prognostic potential in certain settings, its lack of independent significance in multivariate analysis suggests that its role may be secondary to more comprehensive indices such as CAR, which integrate inflammation and nutritional status more effectively.

In MPM, epithelioid histology carries a significant difference in survival with the best prognosis compared to sarcomatoid or biphasic histology [23]. There are many studies in the literature showing the prognostic importance of histologic subtype [24,25,26]. In our study, consistent with the literature, we found that epithelioid histology was an independent prognostic factor associated with longer survival. This finding supports that epithelioid mesothelioma tends to be more sensitive to treatment and has a slower disease progression, which translates into a more favorable prognosis.

The stage at diagnosis is considered to be one of the important predictors of survival in patients with MPM. In studies examining the factors affecting survival in patients with MPM, it is seen that stage is also a prognostic factor [27,28]. Furthermore, according to data from the Dutch Cancer Registry, patients with early-stage MPM had significantly longer survival compared to advanced-stage cases (stage III or IV) where the disease was often metastatic and less responsive to treatment [29]. In our results, stage was found to be a factor affecting survival in both univariate and multivariate analysis. In early-stage disease (stage I or II), where the tumor is localized, patients are eligible for more aggressive treatment options, including surgery and radiotherapy in combination with chemotherapy, thus increasing the efficacy of treatments. Therefore, early detection and staging are critical to improve the clinical outcomes of patients with MPM.

Although the results of our study are consistent with the literature, there are limitations to be considered. First, its retrospective nature may lead to selection bias and limit the generalizability of our findings. Second, data were obtained from patients followed in four different centers, and differences in diagnostic protocols, treatment strategies, and follow-up practices between these institutions may have influenced the results. Third, our study comprises a relatively small number of surgical patients, which may have limited our ability to detect potential survival differences between EPP and PD techniques. Future studies with larger cohorts are needed to validate these findings and better explore survival disparities between different surgical modalities. Fourth, the potential association between CAR and being a surgical candidate has not been analyzed. Although CAR was not used as a criterion for surgical selection in our cohort, we recognize that fitter patients who are more likely to be surgical candidates may exhibit lower CAR values. In addition, there is no standardized cut-off value for inflammatory markers such as CAR and NLR, which may affect their prognostic utility and comparability across studies. Prospective randomized studies with standardized markers and more patients are needed to confirm these findings.

## 5. Conclusions

Clinicopathologic features and inflammatory markers play an important role in the prognosis of MPM. In particular, CAR stands out as an independent prognostic factor as a combined reflection of inflammation and nutritional status. In addition, epithelioid histology and prolonged survival times in patients diagnosed at an early stage demonstrate once again the importance of early diagnosis and appropriate treatment planning. Confirmation of these findings in prospective, multicenter studies and the definition of standard inflammatory marker thresholds may allow for the development of more effective approaches in the management of MPM.

## Figures and Tables

**Figure 1 medicina-61-00144-f001:**
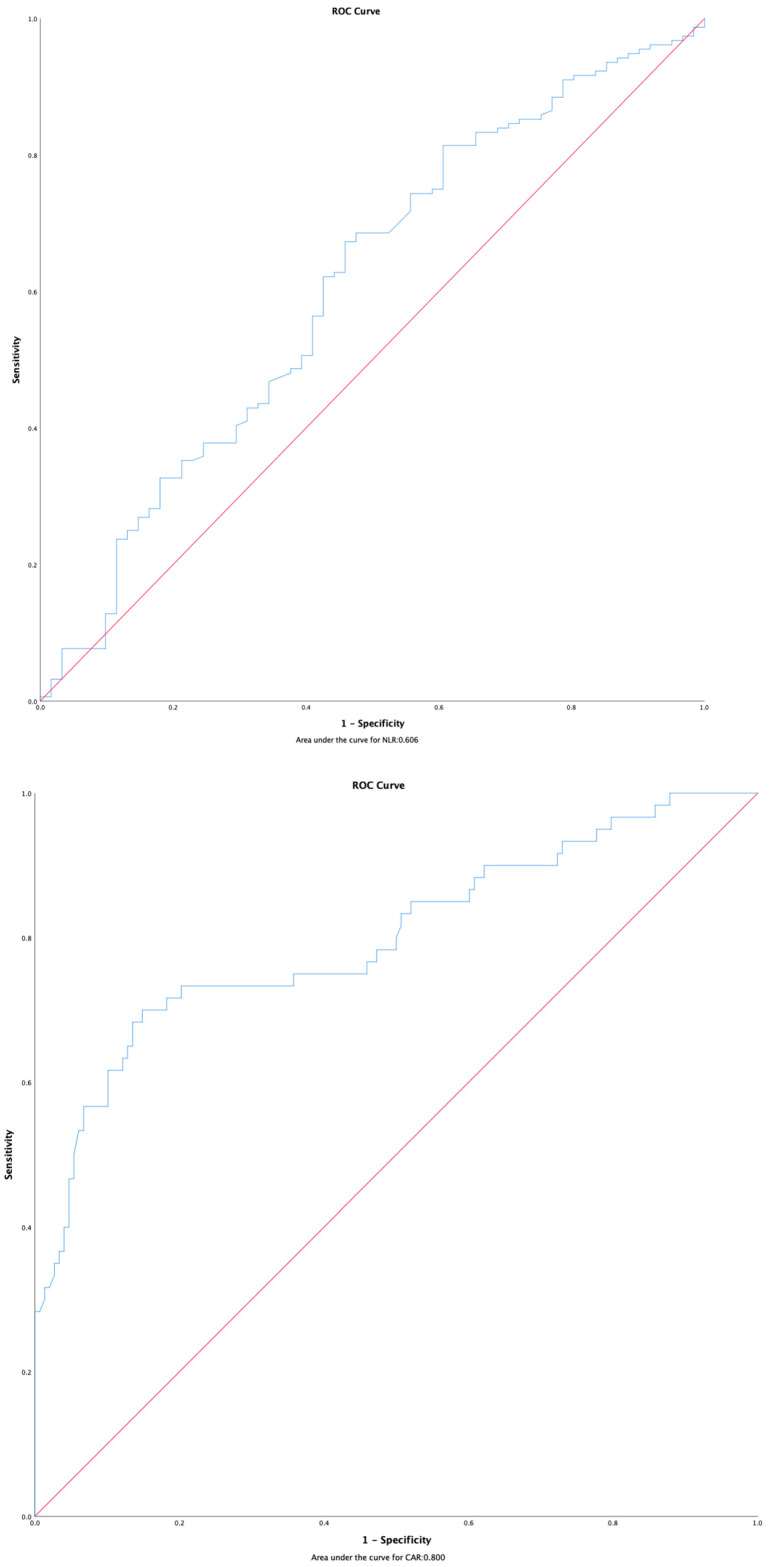
ROC curves of NLR and CAR.

**Figure 2 medicina-61-00144-f002:**
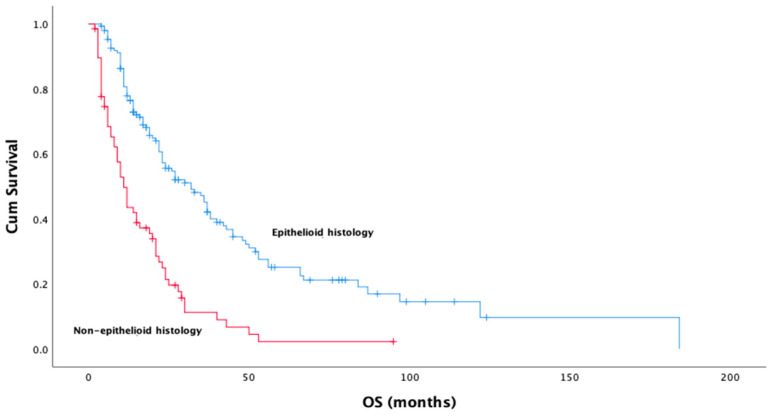
Kaplan–Meier survival analysis by histologic subtype (*p*:0.000).

**Figure 3 medicina-61-00144-f003:**
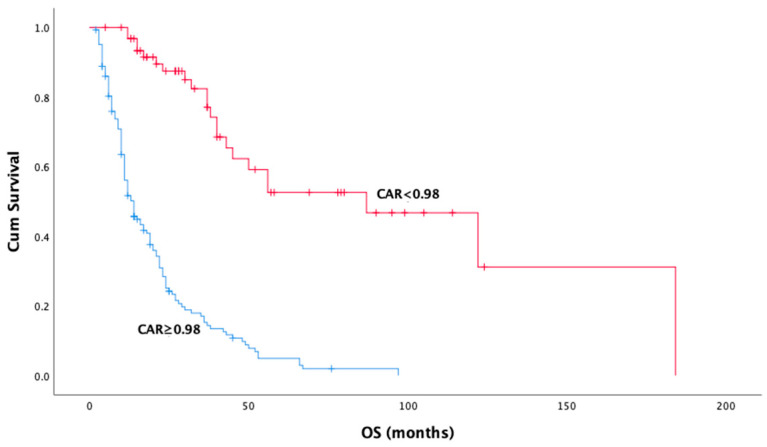
Kaplan–Meier curves for overall survival by C reactive protein/albumin cut off (*p*:0.00).

**Table 1 medicina-61-00144-t001:** Demographic and clinical characteristics of the study patients.

Age, year	
Median (IQR)	59 (52–68.5)
Age group, n (%)	
<60 years	109 (50.2)
≥60 years	108 (49.8)
Gender, n (%)	
Female	77 (35.5)
Male	140 (64.5)
ECOG-PS, n (%)	162 (100)
0–1	192 (88.5)
≥2	22 (11.5)
Histology, n (%)	
Epithelioid	149 (68.7)
Sarcomatoid	37 (17.1)
Biphasic	31 (14.3)
Asbestos exposure, n (%)	
Yes	126 (58.1)
No	90 (41.59
Tobacco exposure, n (%)	
Yes	108 (49.8)
No	109 (50.2)
Most common symptom at presentation, n (%)	
Dyspnea	104 (47.9)
Stage group at diagnosis, n (%)	
Stage I-II-III	136 (62.7)
Stage IV	81 (37.3)
Surgery, n (%)	
Yes	90 (41.47)
No	127 (58.53)
Type of surgery, n (%)	
PD	66 (73.3)
EPP	24 (26.7)
Adjuvant therapy, n (%)	
Yes	78 (86.7)
No	12 (13.3)
Adjuvant radiotherapy, n (%)	
Yes	56 (25.8)
No	161 (74.2)
Recurrence in operated patients, n (%)	
Yes	72 (80.0)
No	18 (20.0)
Systemic treatment, n (%)	
Cisplatin + pemetrexed	147 (67.7)
Carboplatin + pemetrexed	37 (17)
Others	33 (15.3)
Use of immunotherapy in any line, n (%)	
Yes	36 (16.6)
No	181 (83.4)

IQR: Interquartile range; ECOG: Eastern cooperative oncology group; PD: Pleurectomy decortication; Epp: Extrapleural pneumonectomy.

**Table 2 medicina-61-00144-t002:** Clinical and pathological factors related to PFS.

	Univarite	
	Median PFS	*p*
Age		
<60 years	12.88 (8.82–16.93)	0.105
≥60 years	12.22 (5.03–19.40)	
Gender		
Male	10.90 (7.20–14.61)	0.618
Female	13.53 (10.87–16.20)	
Asbestos exposure		
No	12.97 (8.22–17.73)	0.933
Yes	12.22 (8.80–15.64	
Histology		
Epithelioid	14.78 (10.21–19.35)	0.437
Non-epithelioid	9.75 (4.90–14.60)	
Surgery		
EPP	16.42 (7.58–25.27)	0.590
PD	12.22 (9.31–15.13)	
Systemic treatment		
Cisplatin + pemetrexed	14.78 (11.33–18.23)	0.610
Carboplatin + pemetrexed	17.61 (12.38–22.10)	
CRP/Albumin rate		
<0.98	8.96 (5.05–12.88)	0.010
≥0.98	17.61 (13.30–21.91)	
Neutrophil/lymphocyte rate		
<2.58	14.78 (10.21–19.35)	0.550
≥2.58	10.90 (6.75–15.05)	

**Table 3 medicina-61-00144-t003:** Clinical and pathological factors related to OS.

	Univarite		Multivariate	
	Median OS	*p*	HR (%95 CI)	*p*
Age				
<60 years	24.0 (18.77–29.22)	0.020	Ref	0.251
≥60 years	19.0 (12.20–25.79)		1.21 (0.87–1.69)	
Gender				
Male	22.0 (18.26–25.73)	0.535		
Female	23.0 (17.42–28.57)			
Asbestos exposure				
Yes	23.0 (17.14–28.85)	0.820		
No	22.0 (18.02–25.97)			
Histology				
Epithelioid	32.0 (23.51–40.48)	<0.001	Ref	<0.001
Non-epithelioid	11.0 (8.39–13.60)		2.44 (1.72–3.45)	
Metastases at diagnosis				
No	32.0 (24.02–39.97)	<0.001	Ref	<0.001
Yes	12.0 (9.36–14.63)		1.86 (1.33–2.60)	
Surgery				
Yes	36.0 (29.06–42.94)	<0.001	Ref	0.914
No	14.0 (9.83–18.16)		1.02 (0.65–1.61)	
Type of surgery				
PD	37.0 (25.82–48.17)	0.025	Ref	0.956
EPP	30 (16.67–43.32)		1.00 (0.85–1.17)	
Systemic treatment				
Cisplatin + pemetrexed	24.0 (19.52–28.48)	0.455		
Carboplatin + pemetrexed	21.0 (16.64–25.35)			
CT regimen with bevacizumab				
Yes	22.0 (12.73–31.26)	0.995		
No	23.0 (19.63–26.36)			
Use of immunotherapy in any line				
Yes	45.0 (22.03–67.96)	0.016	Ref	0.352
No	21.0 (18.15–23.85)		0.78 (0.46–1.31)	
CRP/Albumin rate				
≥0.98	14.0 (11.02–16.79)	<0.001	Ref	<0.001
<0.98	87.0 (40.32–133.68)		0.17 (0.10–0.28)	
Neutrophil/lymphocyte rate				
<2.58	25.0 (20.37–29.62)		Ref	0.180
≥2.58	21.0 (15.91–26.09)	0.040	1.27 (0.89–1.81)	

OS: Overall survival; HR: hazard ratio; CI: confidence interval; CT: chemotherapy; EPP: extrapleural pneumonectomy; PD: pleurectomy decortication; CRP: C-reactive protein.

## Data Availability

Data is contained within the article.

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
