# Peer review of "The Effect of Systemic Inflammation and Clinicopathologic Features on Survival in Malignant Pleural Mesothelioma: A Multicenter Analysis"

_medicina, 2025, doi:10.3390/medicina61010144_

Round 1
Reviewer 1 Report
Comments and Suggestions for Authors
Dear authors
This is an interesting manuscript. I have a few remarks about your paper:
- row 91 - use mm3 instead of mm3; avoid the space before the )
- row 94 - you need to clarify here - you mention that all your continuous data were presented as median and interquartile range; this is correct for continuous variables with non-normal (non-gaussian) distribution; all your continuous variables were like this? Because the continuous variables with normal (gaussian) distribution are presented as mean and standard deviation;
- in table 1 you present 91 patients who benefited from surgery; however, in the row regarding the type of surgery the sum is 90 (66+24); also the remark regarding recurrence in operated patients - the sum is 90 (72+18); please explain these differences or is it a typo error?
- in table 3 - avoid using values such as 0.000; use <0.001, in the same manner as in the text;
- the Conclusions should be presented as a separate paragraph, like 5. Conclusions, the next paragraph after Discussions
Reviewer 2 Report
Comments and Suggestions for Authors
Dear Editor and Authors,
Thank you for asking me to review this work titled "The Effect of Systemic Inflammation and Clinicopathologic Features on Survival in Malignant Pleural Mesothelioma, A Multicenter Analysis" by Dr. Sever and colleagues from Constantinopole, Turkey.
In this retrospective, single institution analysis the authors collect demographic and lab. based data on 217 patients diagnosed with MPM and calculated derived inflammatory markers such as NLR and PLR. They where able to demonstrate in their unvariate and multivatiate analysis that CAR is a marked associated with MPM!
However, this associated is loosely based and could be due to other factors as well. I have the following comments/queries I would like addressing:
1. How do the authors assess nutritional status as stated in their introduction. Is CAR the only measurement utilized to make this statement?
2. The aim of the study needs to be toned down a bit i.e. "to make important contributions to the development of inflammation-based prognostic models and individualized treatment strategies." Make it more realistic, you are not discovering the "cure for cancer"!!
3. How do you define this as a multicenter study? Referral from other centers for treatment/management/follow up does not constitute a multi-center trial! As such, if patients were assigned care at your institution, then they belong to the final treatment center. This needs to be clarified and edited appropriately where ever mentioned. Did other institutions contribute only data (not actual patients)?
4. Given this is a retrospective study, how complete are the collected data? Are there significant missing data?
5. Extra-pleural pneumonectomy has been abandoned as a therapeutic modality following the MARS trial! Do the authors currently perform EPP? They report an EPP rate of 26.7%!!!
6. The fact that surgical patients were grouped together in the survival analysis with non surgical patients creates an irrevocable bias that needs correction!! How can one assume the two groups/treatment modalities would not have differences in survival and recurrence?
7. In continuation of the above, how did CAR relate to surgery? Why wasn't surgery analyzed as a predictor of survival as it should?? In addition how can the authors exclude that there is no selection bias between been selected for surgery (because one is more fit - i.e. has better CAR) and survival benefit!!
8. In table 2 and in the analysis EPP and PD need to be analyzed as separate variables and not grouped together as "surgery" because it has been shown that the two techniques have a survival disparity between them!!
9. The fact that surgery provides no apparent benefit lead one to consider that the sample size of the patients was too small to show statistical different variations. Was a power analysis performed post hoc?
10. Actual p-values need to be given and not only 0.00 ?
11. Considering immunotherapy is a recently introduced treatment modality for MPM what was the percentage of patients receiving it and from what year onward!!
In conclusion, this study is really not focused!! It analyzes multiple parameters (epidemiological, histological, treatment, demographic) which can be unrelated between them in terms of association with MPM survival!! It does not focus on its main hypothesis/investigating aim (i.e. the role of inflammatory markers only!!) but mixes into it other, well investigated and reported in the literature parameters about MPM!! In addition, I feel it is underpowered / has a small sample size so as to show differences. Therefore, I am having a hard time to support the publication of this work at its current form!!
Round 2
Reviewer 1 Report
Comments and Suggestions for Authors
Dear authors
Thank you fo the revisions that you have made to the manuscript.
I have no further suggestions.
Reviewer 2 Report
Comments and Suggestions for Authors
Dear Editor and Authors,
Thank you for giving me the opportunity to re-evaluate this revised manuscript. Although I still have my reservations regarding its sample size and applicability as well as its overall usefulness to clinical practice, I do aknowledge that the authors have made a genious effort in revising their work and providing an improved manuscript.
As such, I would recommend its publication with the reservations stated above.
Kind regards to all and Happy New Year.